# Diagnostic Potential of Circulating Tumor Cells, Urinary MicroRNA, and Urinary Cell-Free DNA for Bladder Cancer: A Review

**DOI:** 10.3390/ijms23169148

**Published:** 2022-08-15

**Authors:** Dai Koguchi, Kazumasa Matsumoto, Izuru Shiba, Takahiro Harano, Satoshi Okuda, Kohei Mori, Shuhei Hirano, Kazuki Kitajima, Masaomi Ikeda, Masatsugu Iwamura

**Affiliations:** Department of Urology, Kitasato University School of Medicine, 1-15-1 Kitasato Minami-ku Sagamihara, Sagamihara 252-0374, Kanagawa, Japan

**Keywords:** bladder cancer, circulating tumor cells, microRNA, cell-free DNA

## Abstract

Early detection of primary bladder cancer (BCa) is vital, because stage and grade have been generally accepted not only as categorical but also as prognostic factors in patients with BCa. The widely accepted screening methods for BCa, cystoscopy and urine cytology, have unsatisfactory diagnostic accuracy, with high rates of false negatives, especially for flat-type BCa with cystoscopy and for low-risk disease with urine cytology. Currently, liquid biopsy has attracted much attention as being compensatory for that limited diagnostic power. In this review, we survey the literature on liquid biopsy for the detection of BCa, focusing on circulating tumor cells (CTCs), urinary cell-free DNA (ucfDNA), and urinary microRNA (umiRNA). In diagnostic terms, CTCs and umiRNA are determined by quantitative analysis, and ucfDNA relies on finding genetic and epigenetic changes. The ideal biomarkers should be highly sensitive in detecting BCa. Currently, CTCs produce an unfavorable result; however, umiRNA and ucfDNA, especially when analyzed using a panel of genes, produce promising results. However, given the small cohort size in most studies, no conclusions can yet be drawn about liquid biopsy’s immediate application to clinical practice. Further large studies to validate the diagnostic value of liquid biopsy for clinical use are mandatory.

## 1. Introduction

The worldwide absolute incidence of urothelial cancer situates it as the sixth most common cancer in men and the 17th in women, with bladder cancer (BCa) being dominant, accounting for 90% of all urothelial cancers [1,2]. At initial diagnosis, approximately 75% of BCa cases are non-muscle-invasive (NMIBC), and 20% are muscle invasive (MIBC) [3]. Proteomic research, combined with pathology data, has been extensively applied in an attempt to clarify the oncologic characteristics of BCa [4,5,6,7]. Subsequently, molecular analyses have shown large differences in the aggressiveness of NMIBC, and early detection and treatment are known to contribute to favorable oncologic outcomes in a proportion of NMIBC cases [8,9,10]. For MIBC, radical cystectomy has been the standard of care since the early 1990s, and a surgical wait time of less than 3 months has a significant effect on prognosis [11]. Accurate diagnostic tools are, thus, mandatory to achieve a better prognosis for patients with BCa.

The “gold-standard” methods for the detection of BCa are cystoscopy and urine cytology. However, cystoscopy with white light is an operator-dependent procedure with a drawback of low sensitivity (SN), especially for flat-type BCa such as carcinoma in situ [12]. Moreover, the procedure is invasive and sometimes leads to dysuria and urinary tract infection. In contrast, urine cytology is less invasive and has high specificity (SP, 98%) for the detection of BCa, but its SN, especially for low-risk disease, is poor (about 30% at most) [13]. To overcome the resulting high rates of false negatives, several urinary biomarkers have been proposed and are currently testable using commercial kits based on sediment cells or proteins. Nevertheless, results from these kits provide inadequate SN, especially for low-risk BCa. Moreover, their SPs are lower than that for urine cytology, and they are unable to replace the conventional diagnostic methods of cystoscopy and urine cytology for the detection of BCa [13].

The ideal biomarker for diagnosing BCa should be minimally invasive to obtain and should produce high-accuracy results. Since the early 2010s, liquid biopsy has evoked huge interest not only because of easy access to samples, but also because of the abundance of molecular and genomic features of malignancy that can be detected in a sample. Biomarkers potentially available to liquid biopsy are circulating tumor cells (CTCs) and circulating nucleic acids such as cell-free DNA (cfDNA) and microRNA (miRNA) that exist freely or within extracellular vesicles shed mainly by tumor cells into bodily fluids [14]. Although CTCs are obtained from blood samples, cfDNA and miRNA can be evaluated in either blood or urine samples [13]. The nucleic acids in urine samples as opposed to blood samples are more promising for investigation as tools for BCa detection; they are noninvasively obtainable and potentially informative about the genomic features of BCa because of direct contact with urothelial cells. Therefore, in this review, we discuss the diagnostic value of CTCs and of urinary cfDNA (ucfDNA) and miRNA (umiRNA) in BCa.

## 2. Diagnostic Utility of Biomarkers Assessed by Liquid Biopsy

### 2.1. CTCs

Epithelial–mesenchymal transition is believed to endow epithelial cells with a range of mesenchymal characteristics, thereby playing an important role in the multistep process of the hematogenous metastasis of epithelial cancers [15]. Much research has, thus, set out to reveal the underlying mechanism of epithelial–mesenchymal transition, and a possible candidate is CTCs [16]. CTCs detach from a primary site and migrate via the blood and lymphatic systems to distant sites compatible with the growth of the particular CTCs [16]. This “seed and soil” hypothesis firstly proposed by Stephen Paget in 1989 reached general acceptance when emerging technologies were able to isolate CTCs from a patient’s blood [17]. Nevertheless, the theory has remained challenging because CTCs are scarce in blood, accounting for <0.004% of all mononuclear blood cells [18].

Various methods have been developed to isolate CTCs from among the many millions of normal blood cells and to count them [19,20,21]. Historically, those methods have been based mainly on nucleic acids (real-time polymerase chain reaction (PCR) and multiplex real-time PCR) and antigen characteristics (immunocytochemistry, immunofluorescence, and immunofluorescence flow cytometry) [19]. Nucleic acid-based analyses demonstrated especially favorable SN and strong SP by identifying messenger RNA expression of tumor-specific genes such as cytokeratin 20 and epidermal growth factor receptor [20]. However, CTC lysis during the analysis sometimes did not allow for an assessment of cell morphology or further cell analyses [19]. Subsequently, a molecular approach with better reproducibility than that obtained with PCR has currently been adopted as the platform for nearly all CTC-based studies in BCa. In this approach, CTCs are detected using antibodies against the epithelial cell adhesion molecule (EpCAM) antigen, discriminating CTCs with nucleated EpCAM^+^/cytokeratin^+^/leukocyte common antigen 45^−^ from healthy EpCAM^−^/cytokeratin^−^/leukocyte common antigen 45- blood cells [21]. This method is currently the only one that has been approved by the US Food and Drug Administration for monitoring CTCs in patients with metastatic breast, colorectal, and prostate cancer; however, at the time of writing, it had not yet been approved for BCa.

Many studies have nevertheless used this molecular approach to evaluate the diagnostic value of CTCs for BCa. Most were nonrandomized and prospective trials that detected CTCs in patients who were planned to undergo radical cystectomy for either NMIBC or MIBC [22,23,24,25]. One of the largest trials, by Soave et al., found CTCs in 21.3% of 141 patients before surgery [25]. Comparable frequencies of CTCs in patients with BCa, ranging from 18% to 30%, were also reported by other authors [22,23,24]. This low SN is in accordance with the findings of a systematic review and meta-analysis published in 2011 that encompassed 30 studies reported during the 2000s that used the molecular approach [20]. Overall, the SN and SP for CTC detection in BCa were 35.1% (95% confidence interval: 32.4% to 38%) and 89.4% (95% confidence interval: 87.2% to 91.3%), respectively.

These diagnostic results for CTCs in BCa require careful interpretation. First, in several studies, some of the patients (ranging from 6.1% to 12.8%) were staged as pT0, indicating that the CTC detection rate would have been higher had the analysis been focused on patients staged pT1 or greater [23,24,25]. Second (and in contrast), the detection rate might have been lower had the blood sampling been performed before transurethral resection of the BCa, because transurethral resection of aggressive BCa can lead to dissemination of CTCs; a rise in CTC count after surgery was found in 52.9% (9/17) of MIBC cases and in 30.8% (4/13) of cases of high-grade disease [26,27]. Third, the most fundamental concern is that the detection power of the molecular approach is limited by its sole focus on EpCAM. By focusing on tumor cells that exhibit epithelial features, it fails to detect subpopulations of CTCs with intermediate or pure mesenchymal features [28].

Taken together, the research into CTCs has not uncovered a reliable method for analyzing CTCs in BCa. Although the SNs reported when using the molecular approach alone are currently unfavorable, combining a CTC analysis with other liquid biopsy approaches might be one solution. For example, a very recent study found that a CTC analysis of circulating endothelial cells showed potential to guide the diagnosis of BCa [29].

### 2.2. umiRNA

Understanding the role of noncoding RNA in malignancy holds great promise. Although noncoding RNA is not translated into protein, it interacts in complex ways with various biologic processes such as gene splicing, nucleotide modification, protein transport, and regulation of gene expression [30]. Three types of noncoding RNA have been posited on the basis of the number of nucleotides, and the type most investigated in cancer, called miRNA, consists of 19–22 nucleotides [30]. Extensive studies have demonstrated that normal cells and cancerous lesions use exosomes to release miRNAs into blood or urine [13]. Interest in circulating miRNAs as noninvasive biomarkers in cancer has, therefore, been increasing, and urine is the bodily fluid most extensively investigated. In general, there are two reasons for the prevalence of umiRNA analyses. First, compared with messenger RNA, miRNA has the advantage of being less vulnerable to RNase in urine mainly because of its greater chain length [13]. Second, that better stability contributes to the reliability of analyses, with miRNA being superior to tissue samples obtained via transurethral resection of the bladder tumor, which often deteriorate [31]. The preference for umiRNA samples is supported by a review article covering about 70 publications; that article showed a concordance in gene characteristics between BCa tissues and urine samples [31]. In contrast, miRNA in plasma has not yet been well studied.

Detecting umiRNA has been a clinical challenge because exosome concentrations in urine are small (<0.01 vol.%) [32]. Detection of umiRNA involves two steps: isolation of the exosome and quantification of the miRNA. Three isolation methods have been accepted: differential ultracentrifugation, immunoaffinity capture, and size exclusion chromatography. Of the three, differential ultracentrifugation, which enriches particles according to density, is considered the “gold standard” [13]. However, the technique has been reported to have an unfavorable exosome recovery rate of 5% at most in urine samples. Moreover, the abundance of protein in urine lowers the SN of extracellular vesicles because polymeric Tamm–Horsfall proteins and albumin are co-isolated as contaminants in the centrifugation procedure [33], emphasizing the need for a reliable proteomic approach to biomarker discovery in BCa.

In terms of quantification, the first study of umiRNAs in BCa used quantitative real-time PCR to analyze samples from 83 patients [34]. The advent of high-throughput microarray technology then allowed hundreds of miRNAs to be quantified simultaneously. Those two methods are now those most commonly applied in clinical practice, with microarray probably being the most logical first step and quantitative real-time PCR often being used to validate the microarray results or to screen fewer miRNAs across larger numbers of culture conditions [35]. However, these microarray- and PCR-based technologies cannot detect an exhaustive number of miRNAs; thus, umiRNA profiling by next-generation sequencing (NGS) techniques has recently been a key breakthrough. NGS potentially provides a more comprehensive miRNA analysis, including the detection of new miRNAs with tissue-specific expression [35].

Using conventional methods such as microarray- and PCR-based technologies, numerous studies have evaluated the diagnostic role of umiRNA in BCa (Table 1). A small study including six BCa patients and three healthy volunteers showed overexpression of umiR-21-5p with 72.2% SN and 95.8% SP for detecting the disease. In the same cohort, the umiR-21-5p SN was much better than urine cytology (44.4% SN, 100% SP). Intriguingly, the diagnostic accuracy of umiR-21-5p in patients with BCa remained even without positive cytology (75.0% SN, 95.8% SP), indicating that umiR-21-5p might be a biomarker for the detection of early BCa [36]. Higher sensitivity of umiR-21-5p was also reported in a larger study conducted by Ghorbanmehr et al. In 45 patients with BCa and 20 healthy participants, umiR-21-5p had 84% SN and 59% SP, with an area under the curve (AUC) of 0.76 for discriminating BCa from non-BCa [37]. In the analysis, umiR-141-3p and umiR-205-5p were revealed to have diagnostic accuracies lower than that of umiR-21-5p. Notably, the same three umiRNAs were also found to be overexpressed in patients with prostate cancer compared with healthy study participants [37].

To overcome the relatively lower SN of umiRNAs for the detection of BCa, some studies developed combination tests using multiple umiRNAs for improved accuracy [38,41,47,48]. Hofbauer et al. identified a subset of six umiRNAs (let-7c, miR-135a, miR-135b, miR-148a, miR-204, and miR-345) associated with a favorable diagnostic accuracy (88.3% SN) [41]. Mengual et al. identified a different panel of six umiRNAs (miR-187, miR-18a*, miR-25, miR-142-3p, miR-140-5p, and miR-204) that had 84.8% SN and 86.5% SP (AUC 0.92) for diagnosing BCa [38]. This subset was the most accurate in identifying advanced disease with high SP (high-risk NMIBC: 90.3% SN, 86.5% SP; MIBC: 87.1% SN, 86.5% SP) [38]. Given that the two studies achieved comparable results with six different umiRNAs per panel, the most appropriate combination of umiRNAs should be urgently explored for clinical application. In fact, although Urquidi et al. showed an excellent AUC of 0.98 with a 25-umiRNA panel for diagnosing BCa, even a 10-umiRNA model retained a high AUC of 0.90 in the same cohort [47].

One umiRNA that most overlapped in the studies of BCa was umiR-146 [40,42,49]. However, although umiR-146 was found to carry diagnostic value in grade differentiation, the opposite association has also been reported; Baumgart et al. found more overexpression in grade 3 than in grade 2 or lower disease, while Andreu et al. found overabundance in low-grade rather than in high-grade disease [40,42]. This discordance might reflect the inflammatory status in BCa. In a unique cohort recruited by Mearini et al., expression of umiR-146 was increased in patients with BCa compared with their healthy counterparts, but no significant difference was observed when patients with BCa and patients with histologically confirmed bladder inflammation were compared [49]. Using in vitro methods, those authors also showed that umiR-146 is an inflammasome in BCa, targeting a large complex of NOD-like receptors. In fact, umiR-146 was associated with the severity of immunoglobulin A nephropathy, in which the main symptom is generally hematuria, and the studies from Baumgart et al. and Andreu et al. did not mention the presence of hematuria in their cohorts [50]. Further large investigations involving patients with matched inflammatory backgrounds are needed to elucidate the potential diagnostic utility of umiR-146 in BCa.

With respect to T stage, aberrant expression of some umiRNAs has been associated with disease invasiveness. De Long et al. demonstrated overexpression of umiR-940 in patients with MIBC compared with patients with NMIBC and with healthy control participants [39]. Expression of umiR-940 was the highest in advanced disease (pT1 grade 3 and ≥pT2) and the lowest in the absence of malignancy (healthy participants with and without a prior history of urothelial carcinoma). Likewise, expression levels of umiR-26a were higher in high-grade NMIBC and MIBC than in low-grade NMIBC. In contrast, Baumgart et al. also showed an association between downregulation of umiRNA and advanced T stage. Expression of umiRNA-138-5p was observed to be T stage-specific, dropping significantly lower as the T stage became more advanced [42] (fold change value of 0.163). Another two studies of miR-138-5p also showed lower expression in BCa tissues than in normal tissues [51,52]. The potential tumor suppressor role of miR-138-5p in BCa might be explained by its correlation with the epithelial–mesenchymal transition-associated protein encoded by *ZEB2,* which was demonstrated in a transwell cell invasion assay and a scratch wound healing assay [51].

NGS has attracted considerable attention because of its up-to-date deep sequencing technology. Nevertheless, just a few studies have used NGS to analyze umiRNA in BCa. The first publication, from Pardini et al. in 2018, reported altered expression of umiRNAs by tumor stage and grade [43], leading to the design of diagnostic models for three types of BCa. The models included clinical information (age and smoking status) and molecular information (miR-30a-5p, miR-486-5p, and let-7c-5p). The resulting diagnostic accuracies were favorable; the AUCs for NMIBC grade 2 or lower, NMIBC grade 3, and MIBC were 0.73, 0.95, and 0.99, respectively.

Use of NGS to analyze umiRNAs has another advantage, in that researchers can verify their data by referring to published databases such as The Cancer Genome Atlas (TCGA), which generates and analyzes NGS data from the whole genome in various cancer types. Two recent studies explored expression levels of umiRNAs in patients with BCa and, using data from TCGA, validated an overlapping of certain miRNAs with altered expression levels [44,45]. Braicu et al. identified five umiRNAs in common between a patient cohort and TCGA data [44]. Furthermore, using the AmpliSeq Cancer Panel kit (Illumina, San Diego, CA, USA) and Ion PGM Dx Torrent Suite (Thermo Fisher Scientific, Waltham, MA, USA), they further revealed that the umiRNAs with altered expression were associated with certain genes, including *TP53*, *FGFR3*, *KDR*, *PIK3CA,* and *ATM.* Likewise, Lin et al. found four umiRNAs with altered expression in patients with BCa, and subsequent pathway enrichment analysis showed potential target pathways: MAPK, PI3K/AKT, focal adhesion, and Erb [45].

A unique method in addition to NGS data for the diagnosis of BCa was reported by Moisoiu et al. in 2022 [46]. Using surface-enhanced Raman spectroscopy, which amplifies the Raman signal of molecules adsorbed from urine, they discriminated patients with BCa from healthy volunteers. In BCa, a panel of the top three differentially expressed umiRNAs (miR-34a-5p, -205-3p, and -210-3p) found by NGS was combined with the surface-enhanced Raman spectroscopy data, revealing superior diagnostic accuracy for the combination. In the NGS-based studies described earlier, the umiRNAs with altered expression in BCa were not consistent; however, three of the four studies found an increased umiRNA in common: miR-205-5p [43,44,46]. Braicu et al. performed Ingenuity Pathway Analysis and found that epithelial–mesenchymal transition appears to be associated with the miR-205-5p function. However, those studies analyzed only 20–116 samples; further large studies will be of great importance in identifying genetic markers in BCa, including those for stage and grade, helping to facilitate precision medicine in the disease [43,44,45,46].

### 2.3. Urine Cell-Free DNA

Fragmented cfDNA is presumed to derive from cancer cells that have undergone necrosis and apoptosis; it is usually found as a double-stranded structure [53]. Profiling such DNA is expected to allow for the identification of genetic heterogeneity in malignancy. cfDNA in plasma has been extensively researched in various cancers since Jahr et al. first discovered its presence in the early 1940s [54,55]. Today, interest in ucfDNA in BCa is growing because of the tumor’s close contact with urine and the less invasive means needed for obtaining samples. That interest has been rewarded by the finding of a higher concordance between gene mutations in urine and BCa tissues than in plasma and tissues [56].

As yet, no standardized method for isolating ucfDNA has been established. The relatively low concentration and mostly short fragments of ucfDNA make it challenging to find assays with high SN and reproducibility. Multiple methods have, therefore, been introduced in research papers, and ucfDNA can be isolated using either commercial kits or classical laboratory techniques [57]. Currently, researchers tend to favor commercial kits, which are able to isolate low-molecular-weight DNA [58].

A common strategy for detecting ucfDNA uses a whole-genome sequencing technique such as digital PCR and NGS [58]. Both techniques can detect rare mutations in cancer, but the approaches are different. Digital PCR has the potential to detect point mutations in low allele fractions [59,60]. Two popular platforms are droplet digital PCR and BEAMing (beads, emulsions, amplification, and magnetics). In contrast, although the SN for rare mutations is much lower with NGS (≤2%), a wide range of genomic alternations can be detected without prior tumor sequencing, thus helping to identify rare mutations and to detect primary cancer [58]. Furthermore, recent advances in the detection and analysis of ucfDNA by NGS have improved the error rate to 0.01–0.001% from 0.1–1% [59,61,62]. These novel assays are expected to offer deeper insights into the clinical utility of ucfDNA in BCa.

A variety of methods using ucfDNA for the diagnosis of BCa have been reported (Table 2). Some studies used a simple quantitative method with real-time PCR. Brisuda et al. showed that an increased concentration of ucfDNA in second urine (voided 2–3 h after the first morning urine) had an AUC of 0.73 with low SN (42.4% overall; 20.7% for low-grade disease and 59.5% for high-grade disease) in discriminating patients with BCa from healthy participants [63]. Higher SN was reported by Kim et al., who showed diagnostic value for the overexpression of topoisomerase IIα in detecting BCa, with levels increasing as the stage of the disease advanced (70.1% SN for NMIBC and 88.2% SN for MIBC). Nevertheless, the SP was less than 75% at both stages, an unfavorable false-negative rate that might reflect the inclusion of patients with hematuria in the cohort [64].

Using NGS to target the most commonly mutated region in ucfDNA has been the main approach in attempting to accurately detect BCa [67,68]. Dudley et al. achieved acceptable diagnostic accuracy with mutation in *PLEKHS1* for BCa in its early stages, mainly NMIBC (Ta: 76%; T1: 9%) [67]. A large study by Descotes et al., including 348 patients with BCa and 167 healthy control participants [68], focused on mutation in *TERT* because increased *TERT* activity generally leads to elimination of cancer. Mutation in *TERT,* the most frequently reported gene in the literature, is accordingly presumed to be a representative marker of carcinogenesis [56,65,69]. In BCa this mutation had 80.5% SN and 89.8% SP, where urine cytology had only 33.6% SN in the same cohort. Furthermore, the mutation was stage-independent, albeit with a significantly higher frequency in high-grade compared with low-grade disease (84.8% vs. 74.3%, *p* < 0.015).

Mutation in *FGFR3* was also commonly assessed in ucfDNA in various studies [56,65,69]. When seeking a suitable panel to achieve diagnostic accuracy in BCa superior to that achieved with a single marker, preliminary surveillance often made *TERT* the most frequent choice and *FGFR3* the second. Ou et al. reported *TERT* mutations at 46% and *FGFR3* mutations at 38%, while Kissel et al. reported *TERT* at 73% and *FGFR3* at 38% [56,65]. Consistency in the *TERT* positivity rate and discrepancy in the *FGFR3* positivity rate in BCa was attributed to the prevalence of *TERT* and *FGFR3.* Expression of the *TERT* mutation was not stage-dependent, being present in generally high proportions in urine from patients with BCa; the *FGFR3* mutation was more prevalent in patients with NMIBC than in those with MIBC [62]. A panel including those two genes with other common genes demonstrated a notable AUC of 0.94 [56] and a slightly higher AUC of 0.96 when combined with a gene methylation assay [65].

Examining structural alterations in ucfDNA is another way to diagnose BCa. Tumor cells are presumed to release DNA fragments longer than the fragments released by normal cells [58]. Casadio et al. analyzed the potential role of ucfDNA integrity as a diagnostic marker for early BCa [66]. Those authors chose four oncogenic genes longer than 250 bp as determined by real-time PCR and calculated integrity on the basis of cycle threshold. Although the approach was simple and inexpensive, ucfDNA integrity did not prove to provide good accuracy (73% SN, 84% SP) for detecting BCa in symptomatic patients [66]. Another structural feature of ucfDNA called “jagged end” can be diagnostically informative, as highlighted by Zhou et al., who assessed single-stranded ends with 5′ nucleosomal protrusions and showed that this jagged end index was lower in patients with BCa than in healthy participants (AUC 0.83). The jagged ends were associated with DNAse I activity (which plays an important role in the fragmentation of plasma DNA), and Zhou et al. observed a tendency for DNAse I expression to gradually decrease from stage Ⅳ to stage Ⅰ BCa [73]. Although cfDNA is generally more degenerated in urine than in plasma, jagged ends are reportedly more abundant in urine than in plasma [73,75]. The mechanism that preserves that pattern requires further investigation to reach a comprehensive understanding of jagged ends.

Epigenetic change in ucfDNA has also recently attracted attention as a useful method for diagnosing BCa. In particular, focal hypermethylation of certain DNA regions is frequently observed in tumor cells [70]. Although DNA methylation has been detected in urine sediments in various cancer types, Hentschel et al. were the first to report the acceptable performance of urine supernatant as a diagnostic tool in BCa. However, the best diagnostic tool, a *GHSR* plus *MAL* panel using urine sediment, reached an AUC of just 0.87 (78.6% SN, 91.7% SP) [70]. Combined tests of this kind, using two complementary biomarkers selected from among several hypermethylated genes (for example, cg21472506 with cg11437784, and *ONECUT2* with *VIM*), have become increasingly prevalent, showing comparable results across studies using two methylation biomarkers [71,72].

Notably, however, Deng et al. developed a single methylation biomarker model [74]. They used multiple validation steps involving TCGA data, BCa cell lines, and BCa patient cohorts to find the hypermethylation marker that provided the best diagnostic result; in the end, *DMRTA2* achieved 82.9% SN and 92.5% SP, with an AUC of 0.93 in the diagnosis of BCa. The *DMRTA2* test was even more useful for detecting pT1 and pT2 BCa, achieving up to 92.0% SN. That high accuracy is of great importance, given that a large proportion of patients with organ-confined BCa can potentially undergo radical cystectomy with curative intent. More intriguingly, *DMRTA2* is also highly diagnostic for upper-tract urothelial cancers (renal pelvic cancer: 82.9% SN; ureteral cancer: 52.6% SN), indicating its potential for superiority to urine cytology as a diagnostic marker across all urothelial cancers. The clinical utility of *DMRTA2* is potentially worth validating in large cohorts.

## 3. Conclusions

New biomarkers such as CTCs, umiRNA, and ucfDNA that are expected to be able to replace conventional diagnostic methods should be at least highly sensitive in detecting primary BCa. Otherwise, they would engender a considerable risk of missing clinically significant BCa. Currently, the literature relating to CTCs shows little diagnostic value, but studies relating to umiRNA and ucfDNA, especially when a panel of genes is used, provide promising results. However, challenges remain before umiRNA and ucfDNA can be applied in clinical practice. First, researchers have not yet found the best complementary patterns of gene combinations. The BCa literature contains oddly discrepant overlaps in umiRNA, and the variety of diagnostic methods that use the genetic or epigenetic features of ucfDNA have not yet been narrowed down. Second, almost all studies of umiRNA and ucfDNA did not indicate whether their analyses are specific only for BCa, and some of the candidate genes that have diagnostic value in BCa, such as *FGFR3* and *DMRTA2,* are generally not rare in other cancers. Third, although NMIBC is the dominant BCa at initial diagnosis, the SN achievable with several markers is relatively lower for detecting NMIBC than for detecting MIBC. Overall, most studies were limited to small cohorts of less than 100 patients. Studies in larger cohorts should be performed to validate the diagnostic accuracy of liquid biopsy for clinical application.

## Figures and Tables

**Table 1 ijms-23-09148-t001:** Diagnostic accuracy of urinary microRNAs (miRNAs) for bladder cancer (BCa).

Study	Year	Target (Expression in BCa)	BCa/Ctl (n)	Primary Findings	Ref.
Mengual et al.	2013	Panel of six miRNAs:	181/136	84.8% SN, 86.5% SP; AUC 0.92 (overall) 77.6% SN, 86.5% SP (low-grade NMIBC) 90.3% SN, 86.5% SP (high-grade NMIBC) 87.1% SN, 86.5% SP (MIBC)	[38]
		miR-18a* (↑)			
		miR-25 (↑)			
		miR-140-5p (↓)			
		miR-187 (↑)			
		miR-142-3p (↓)			
		miR-204 (↓)			
De Long et al.	2015	miR-940 (↑)	85/45	pT2 or greater, pT1 grade 3 > pT1 grade 1, Ctl	[39]
		miR-26a (↑)		pT2 or greater > pT1 grade 1; pT1 grade 3 > pT1 grade 1; Ctl > pT1 grade 1	
Matsuzaki et al.	2017	miR-21-5p (↑)	6/3	72.2% SN, 95.8% SP (overall)	[36]
Andreu et al.	2017	miR-146 (↑)	36/9	Low-grade > high-grade	[40]
Ghorbanmehr et al.	2018	miR-21-5p (↑)	45/20	84% SN, 59% SP; AUC 0.76 (overall)	[37]
		miR141-3p (↑)		71% SN, 71% SP; AUC 0.74 (overall)	
		miR205-5p (↑)		82% SN, 62% SP; AUC 0.73 (overall)	
Hofbauer et al.	2018	Panel of six miRNAs:	87/115	AUC 0.88 (overall) AUC 0.88 (low-grade NMIBC) AUC 0.93 (high-grade NMIBC) AUC 0.91 (MIBC)	[41]
		Let-7c (↓)			
		miR-135a (↓)			
		miR-135b (↑)			
		miR-148a (↓)			
		miR-204 (↓)			
		miR-345 (↑)			
Baumgart et al.	2019	miR-146	37/0	grade 3 > grades 1, 2 pTa > pT1 > pT2> pT3–4	[42]
		miR-138-5p			
Next-Generation Sequencing					
Pardini et al.	2018	Panel of three miRNAs:	66/48	AUC 0.70 (overall) AUC 0.73 (low-grade NMIBC) AUC 0.95 (high-grade NMIBC) AUC 0.99 (MIBC)	[43]
		let-7c-5p (↑)			
		miR-30a-5p (↑)			
		miR-486-5p (↓)			
Braicu et al.	2019	miR-141-3p (↑)	23/23	AUC 0.86 (overall) AUC 0.89 (overall) BCa < Ctl BCa < Ctl BCa > Ctl	[44]
		miR-205-5p (↑)			
		miR-139-5p (↓)			
		miR-143-5p (↓)			
		miR-200b-3p (↑)			
Lin et al.	2021	Let-7b-5p (↑)	180/100	BCa > Ctl BCa > Ctl BCa > Ctl BCa > Ctl BCa > Ctl	[45]
		miR-146a-5p (↑)			
		miR-149-5p (↑)			
		miR-193a-5p (↑)			
		miR-423-5p (↑)			
Moisoiu et al.	2022	Panel of three miRNAs:	15/16	AUC 0.84 (miRNA alone) AUC 0.84 (SERS alone) AUC 0.92 (miRNA + SERS)	[46]
		miR-34a-5p (↑)			
		miR-205-5p (↑)			
		miR-210-3p (↑)			

Ctl: healthy control participants; SN: sensitivity; SP: specificity; AUC: area under the curve; NMIBC: non-muscle-invasive bladder cancer; MIBC: muscle-invasive bladder cancer; SERS: surface-enhanced Raman spectroscopy.

**Table 2 ijms-23-09148-t002:** Diagnostic accuracy of urinary cell-free DNA (ucfDNA) for bladder cancer (BCa).

Study	Year	Target (Expression in BCa)	BCa/Ctl (n)	Primary Findings	Ref.
Brisuda et al.	2015	ucfDNA concentration (↑)	66/34	42.4% SN, 91.2% SP, AUC 0.73 (overall)	[63]
Kim et al.	2016	Topoisomerase IIA expression (↑)	83/115	73.8% SN, 68.3% SP, AUC 0.74 (overall) 70.1% SN, 63.3% SP, AUC 0.70 (NMIBC) 88.2% SN, 74.8% SP, AUC 0.84 (MIBC)	[64]
Kessel et al.	2017	Panel of six genes	97/103	93% SN, 86% SP, AUC 0.96 (overall)	[65]
		Mutation:			
		FGFR3 (↑)			
		TERT (↑)			
		HRAS (↑)			
		Methylation:			
		OTX1 (↑)			
		*ONECUT2* (↑)			
		*TWIST1* (↑)			
Casadio et al.	2017	ucfDNA integrity (>250 bp)	46/32	73% SN, 84% SP (overall)	[66]
Dudley et al.	2019	Mutation: *PLEKHS1* (↑)	54/34	84% SN, 96% SP (overall)	[67]
Descotes et al.	2020	Mutation: *TERT* (↑)	348/167	80.5% SN, 89.8% SP (overall) 79.4% SN (pTa) 77.6% SN (pT1) 85.2% SN (MIBC) 84.8% SN (high-grade) 74.3% SN (low-grade)	[68]
Ou et al.	2020	Panel of five genes	92/33	AUC 0.94 (overall) 16–46% SN each 100% SP each	[56]
		Mutation:			
		*TERT* (↑)			
		*FGFR3* (↑)			
		*TP53* (↑)			
		*PIK3CA* (↑)			
		*KRAS* (↑)			
Hayashi et al.	2020	Mutation: *TERT* promoter + *FGFR3* hotspot	74/52	68.9% SN, 100% SP (overall) 85.9% SN (with cytology)	[69]
Hentschel et al.	2020	Methylation: *GHSR* + *MAL*	14/12	78.6% SN, 91.7% SP, AUC 0.87 (overall)	[70]
Chen et al.	2020	Methylation: cg21472506 + cg11437784	109/66	90.0% SN, 83.1% SP (overall)	[71]
Ruan et al.	2021	Methylation: *ONECUT2* + *VIM*	192/98	87.1% SN, 82.9% SP, AUC 0.90 (overall)	[72]
Zhou et al.	2021	ucfDNA jagged ends	43/39	AUC 0.83 (overall)	[73]
Ward et al.	2022	Panel of 23 genes Mutation	443/162	87.3% SN, 84.8% SP (overall) 97.4% SN (grade 3) 86.5% SN (grade 2) 70.8% SN (grade 1)	[62]
Deng et al.	2022	Methylation: *DMRTA2*	44/83	82.9% SN, 92.5% SP, AUC 0.93 (overall) 92.0% SN (pT1, pT2)	[74]

Ctl: healthy control participants; SN: sensitivity; SP: specificity; AUC: area under the curve; NMIBC: non-muscle-invasive bladder cancer; MIBC: muscle-invasive bladder cancer.

## Data Availability

Not applicable.

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
