# Peer review of "Diagnostic Potential of Circulating Tumor Cells, Urinary MicroRNA, and Urinary Cell-Free DNA for Bladder Cancer: A Review"

_ijms, 2022, doi:10.3390/ijms23169148_

Round 1

Reviewer 1 Report

The authors invested effort in gathering the data and composing the manuscript. The result, however, is unreadable. A few examples: 1. The first sentence of the abstract: Early detection of bladder cancer is vital, because the patient’s prognosis is typically stage and grade dependent. While the two parts of the sentence are correct, there is no direct connection between them. 2. The second sentence: The standard screening methods… Sorry but there are no standard screening methods in bladder cancer. 3. In the same sentence the authors combine cytology and cystoscopy and conclude that they suffer from “high rates of false negative”. 4. The first reference should provide information on the prevalence of bladder cancer, yet it is referring to two case reports of superficial bladder cancer metastatic to the lungs.

Author Response

  1. The authors invested effort in gathering the data and composing the manuscript. The result, however, is unreadable. A few examples: 1. The first sentence of the abstract: Early detection of bladder cancer is vital, because the patient’s prognosis is typically stage and grade dependent. While the two parts of the sentence are correct, there is no direct connection between them.

Answer: Thank you very much for your useful comment and understanding of the two parts of the sentence. As you pointed out, the first sentence of the abstract was ambiguous, and we corrected the two parts as below.

Page1, “Abstract” paragraph1, line 11-13.

Early detection of primary bladder cancer (BCa) is vital, because stage and grade have been generally accepted not only as categorical but also as prognostic factors in the patients with BCa.   

#2. The second sentence: The standard screening methods… Sorry but there are no standard screening methods in bladder cancer.

Answer: Thank you very much for sharp opinion. We should have paid more attention” to the use of the word “standard” in writing our manuscript. We accordingly corrected it to “widely accepted” in the manuscript.

Page1, “Abstract” paragraph1, line 13.

The widely accepted methods for the diagnosis of BCa, cystoscopy and urine cytology, have unsatisfactory diagnostic accuracy,

#3. In the same sentence the authors combine cytology and cystoscopy and conclude that they suffer from “high rates of false negative”.

Answer: I appreciate reviewer’s comment on “high rates of false negative”. We agree with you that the word did not accurately reflect the cumulative evidence of the diagnostic feature of urine cytology. Then we refer it more specific in the manuscript as below.

Page1, “Abstract” paragraph1, line 13-15.

The widely accepted methods for the diagnosis of BCa, cystoscopy and urine cytology, have unsatisfactory diagnostic accuracy, with high rates of false negatives especially for flat-type BCa with cystoscopy and for the low-risk disease with urine cytology.

#4. The first reference should provide information on the prevalence of bladder cancer, yet it is referring to two case reports of superficial bladder cancer metastatic to the lungs.

Answer: We appreciate your kind advise and subsequently replace the ex-reference no.1 with a new one, which is more suitable for the sentence.

Page11, “Reference”, line 394-395.

Ploeg, M.; Aben, K.K.; Kiemeney, L.A. The present and future burden of urinary bladder cancer in the world. World J Urol. 2009, 27, 289–293.

Reviewer 2 Report

The manuscript submitted for review titled. "Diagnostic Potential of Circulating Tumor Cells, Urinary Mi-2 croRNA and Urinary Cell-Free DNA for Bladder Cancer" presents a review of the recent literature on the molecular diagnosis of bladder cancer. 

The article is very well written and can be an excellent source of knowledge on the topic presented, but the authors should make some minor improvements to the article:

1. I think two additional articles on microRNA biomarkers should also be mentioned in the paper: doi: 10.18632/oncotarget.13382 and doi: 10.18632/oncotarget.16586

2. To broaden the usefulness of the review, I would also suggest adding molecular markers such as circulating mRNAs detected in urine.

Author Response

  1. I think two additional articles on microRNA biomarkers should also be mentioned in the paper: doi: 10.18632/oncotarget.13382 and doi: 10.18632/oncotarget.16586.

Answer: We appreciate reviewer’s suggestion and added the two references in the manuscript (doi: 10.18632/oncotarget.13382→ref no.40 and doi: 10.18632/oncotarget.16586→ref no.41) and more specifically referred to “doi: 10.18632/oncotarget.13382”. Then, ex-reference no.40 and greater than no.40 were corrected to new reference number, which was added two to the ex-one in the manuscript and the tables.  

Page5, “umiRNA” paragraph1, line186.

Some studies developed combination tests using multiple umiRNAs for improved accuracy [38, 39, 40, 41].

Page5, “umiRNA” paragraph5, line194-196.

Given that the two studies achieved comparable results with 6 different umiRNAs per panel, the most appropriate combination of umiRNAs should be urgently explored for clinical application. In fact, although Urquidi et al. showed an excellent AUC of 0.98 with a 25-umiRNA panel for diagnosing BCa, even a 10-umiRNA model remained a high AUC of 0.90 in the same cohort [40].

Page12 “Reference”, line479-483.

  1. Urquidi, V.; Netherton, M.; Gomes-Giacoia, E.; Serie, D. J.; Eckel-Passow, J.; Rosser, C. J.; Goodison, S. A microRNA biomarker panel for the non-invasive detection of bladder cancer. Oncotarget 2016, 7, 86290-86299.
  2. Du, L.; Jiang, X.; Duan, W.; Wang, R.; Wang, l.; Zheng, G.; Yan, K.; Wang, L.; Li, J.; Zhang, X.; et al. Cell-free microRNA expression signatures in urine serve as novel noninvasive biomarkers for diagnosis and recurrence prediction of bladder cancer. Oncotarget 2017; 8: 40832-40842.

#2. To broaden the usefulness of the review, I would also suggest adding molecular markers such as circulating mRNAs detected in urine.

Answer: We appreciate your intriguingly advice. As you pointed out, mRNAs have also been highlighted as potentially diagnostic biomarker for bladder cancer among literatures. I am afraid, however, that the number of words of our manuscript have already reached more than 4000 words and put 2 tables, and there is subsequently not enough space to wright about mRNAs. Additionally, we mentioned mRNAs in the “umiRNA section” in the manuscript, implying a possibility that umiRNAs might be more suitable for the diagnosis of bladder cancer than mRNAs because of its stability against RNase in urine. Therefore, we would like to focus on CTCs, umiRNAs and ucfDNA in this manuscript. Thank you again for your interest and we will discuss this topic in the next paper.